# Time on Their Side: How Visual Timers Affect Anticipatory Anxiety, Performance, and On-Task Behavior in Elementary Math Assessments

**DOI:** 10.3390/ejihpe15120243

**Published:** 2025-11-28

**Authors:** Quentin Hallez, Victoire Vallier

**Affiliations:** Unité de Recherche DIPHE “Développement, Individu, Processus, Handicap et Éducation”, Institut de Psychologie, University Lumière Lyon 2, 69500 Bron, France; victoire.vallier@yahoo.fr

**Keywords:** visual timer, assessment, behavior, anxiety, children

## Abstract

This study investigated the impact of visual timers on 7- to 9-year-old students’ mathematical performance, anticipatory anxiety related to math assessment, and on-task behavior during a timed assessment. Building on previous findings that visual timers reduce anxiety and improve performance, this research further examined how children interact with a Time-Timer and whether its use influenced off-task behaviors. Forty-four children completed a timed mathematical assessment under two counterbalanced conditions: with and without a visible Time-Timer. Results replicated the anxiolytic effect of visual timers indicating significantly lower anticipatory anxiety levels prior to task onset in the Time-Timer condition. However, no significant difference in math performance was observed between conditions. Notably, the Time-Timer condition was associated with a significant reduction in inattentive and motor instability behaviors. This effect was particularly pronounced for children at higher risk for Attention Deficit Hyperactivity Disorder, as measured by the Conners’ questionnaire. Furthermore, engagement with the Time-Timer was highly heterogeneous; while some children never consulted the device, 25% of participants monitored it with high frequency (i.e., more than seven times in a five-minute period). While this study supports the use of visual timers for reducing anxiety and promoting on-task behaviors, it highlights the need to understand individual differences in usage.

## 1. Introduction

For many children, the abstract concept of time can be a source of frustration and anxiety, particularly in the structured environment of an assessment. The rigid temporal structure of assessments, characterized by a clearly defined beginning, middle, and end, can have a significant impact on students’ cognitive abilities. To address this challenge, visual timers are becoming increasingly common in classrooms and therapy settings. They help children visualize and manage their time ([23]; [21]; [34]). This is especially relevant given that children’s perception of time develops gradually with age ([13], [14]), and tools that make time concrete can facilitate this process ([42]; [55]). By supporting the development of time perception and management, visuals timers can play a crucial role in school settings.

[28] ([28]) demonstrated that the presence of a visual timer during a math assessment reduced children’s assessment-related anxiety and improved performance. Performance improvements were particularly striking in students having strong attentional skills. However, although their study highlighted the potential benefits of visual timers, [28] ([28]) did not specifically analyze whether and to what extent children looked at and used the visual timer during the task. Nor did they investigate whether timer use was associated with a reduction in off-task behaviors. Building on this work, the present study aims to replicate these findings while also examining how children interact with the visual timer, an analysis crucial for developing more equitable and effective classroom practices. The findings could enhance our understanding of children’s time management under evaluative conditions and inform pedagogical practices aimed at supporting both academic performance and well-being.

### 1.1. Keeping Track of Time and Attention Capacities

Research over recent decades has demonstrated that humans, like all animals, possess an internal clock system that allows them to measure time. However, while our brains have built-in mechanisms for tracking time, this system does not always lead to accurate time estimation. Humans often perceive time as either expanded or contracted, judging it to be longer or shorter than it actually is. This distortion in time perception is influenced by several factors, such as the attention allocated to processing time ([44]). When our attention is diverted from the passage of time, we tend to perceive durations as shorter. This tendency to underestimate time when distracted has been widely documented in behavioral studies using the dual-task paradigm ([5]; [6], [7]; [9]; [30]; for a review, see [4]). In this paradigm, participants judge the duration of a stimulus either alone or while simultaneously performing a secondary, non-temporal task (e.g., calculation, color discrimination, or memory task). Regardless of the specific secondary task used, results consistently show that perceived duration is shorter in the dual-task condition compared to the single-task condition.

For instance, [25] ([25], [26]) found that 5- to 8-year-old children’s performance on both time reproduction and color discrimination tasks declined when performed simultaneously. Interestingly, authors also showed that this bias decreases with increasing selective attention abilities. Selective attention allows individuals to prioritize stimuli that are relevant to their own needs and goals, filtering out distractions and enabling them to focus on what matters most ([35]; [56]). This suggests that time processing, in the context of an evaluation, competes for attentional resources, creating a cognitive load that can negatively impact performance on both temporal and non-temporal tasks ([25], [26]; [33]).

Since attention plays a fundamental role in time estimation, researchers have therefore formalized an attentional mechanism into internal clock models. According to the pacemaker-accumulator model ([22]), all individuals possess an “internal clock” that allows them to process time, composed of a pacemaker, a switch, and an accumulator. This theoretical model posits that time perception relies on a three-component system. First, an internal pacemaker generates neural pulses at a constant rate, serving as the raw units of subjective time. Second, an attentional switch regulates the transmission of these pulses to the next stage. When attention is directed towards monitoring duration, the switch opens. Conversely, when attention is absorbed by a non-temporal task or stimulus, the switch tends to close, blocking the pulses. Finally, an accumulator sums the total number of pulses that have passed through the switch, with this total forming the basis for our internal representation of elapsed time. This mechanism explains how engaging in a task can lead to an underestimation of duration, as attention is diverted from timekeeping.

Indeed, the perceived duration directly depends on the number of stored pulses: more pulses meaning a longer perceived duration. A decrease in attention to time can affect the “switch” by causing it to flicker between open and closed states during the timing process. This “flickering” switch” phenomenon means that the switch intermittently fails to remain open, preventing some pulses from the pacemaker from reaching the accumulator. Consequently, fewer pulses are stored, leading to an underestimation of the elapsed time.

This explains why distractions can make time seem to fly by and why individuals with attention deficits, such as those with Attention-Deficit/Hyperactivity Disorder, often struggle with time estimation ([41]). Evaluating behaviors related to attentional and executive functions therefore allows us to examine whether the visual timer can act as an external support for temporal regulation and exert an influence. Moreover, the use of a visual timer could be a way to compensate for inequalities in evaluation situations. More recent neurocomputational models further emphasize the central role of attention in time perception, suggesting that it is not just a simple on/off switch but a complex process that integrates temporal information with other cognitive functions ([29]). Therefore, the cognitive load associated with simultaneously processing time while performing a mathematical task reduces performance on the mathematical task ([28]). In a timed test without a visual timer, the participant relies on their internal clock, which consumes cognitive resources in addition to the task. This study hypothesizes that the presence of a visual timer would free up cognitive space and therefore improve performance. In addition, it would reduce dependence on the internal clock. Given these attentional difficulties, a visual timer could help reduce off-task behaviors (such as motor instability and inattention) and provide support for temporal organization during a task.

### 1.2. Evaluation Anxiety

In close connection with attentional skills, it is also relevant to consider the concept of evaluation anxiety, particularly its anticipatory form. This refers to the anxiety experienced in evaluative situations ([58]). Zeidner identifies two components of evaluation anxiety: a cognitive component (worrying about one’s own performance) and an emotional component (physiological arousal—e.g., sweaty palms, racing heart—and negative emotions). The phenomenon of performance decline among individuals experiencing evaluation anxiety is well-documented in university and high school students ([20]; [48]).

The cognitive mechanisms involved in evaluation anxiety appear to center on attention, as described in the cognition–attention interference theory ([51]) and, more recently, in the attentional control theory ([19]). These theories propose that deficits in attentional control play a central role in the development of anxiety. A 2019 meta-analysis by ([52]), which included 58 studies, confirmed the link between anxiety levels and deficits in attentional control. According to this hypothesis, individuals experiencing evaluation anxiety divide their attention between the task at hand and distracting thoughts of doubt, worry, and self-criticism. This divided attention reduces focus on the task and leads to poorer performance.

In this context, time constraints in assessments are a factor to consider, especially since they are widely adopted in education systems worldwide, with the Program for International Student Assessment (PISA) being one example among many, which imposes time limits to ensure standardization and comparability across countries ([45]). However, these constraints are known to exacerbate assessment anxiety ([58]), a finding confirmed by international studies ([46]; [54]). Thus, while time limits ensure the implementation of more easily comparable and standardized assessments, they may also intensify attentional disruptions and further impair the performance of anxious students.

This aligns with the findings of [28] ([28]), who observed that time-constrained assessments without a visual timer resulted in higher evaluation anxiety and lower math scores among participants. However, this study suggests that the use of a visual timer may help reduce the cognitive load associated with temporal processing before task initiation, thereby allowing children to better regulate their anxiety in a time-constrained context.

### 1.3. The Role of Executive Functions and On-Task Behaviors

The positive impact of the visual timer observed in [28]’s ([28]) study may be partly attributed to a general effect on executive functions. Executive functions encompass a set of higher-order cognitive processes that enable goal-directed behavior, including planning (the ability to create a roadmap to reach a goal or to complete a task), working memory (the ability to hold and manipulate information in mind over short periods), inhibition (the ability to control one’s attention, behavior, thoughts, and/or emotions to override a strong internal predisposition or external lure), and cognitive flexibility (the capacity to switch gears and adjust to changed demands, priorities, ([11], [12]). In the context of a timed assessment, these functions are crucial for effectively managing time, maintaining focus, and regulating emotions.

Executive functions contribute not only to cognitive control but also to the regulation of anticipatory anxiety. This is a central characteristic of childhood anxiety disorders, identified by increased emotional reactivity to uncertainty and novel situations ([57]). The increased amygdala activation observed during uncertain anticipation suggests that children experience anxiety even before the task begins. Given this interaction between executive control and anticipatory anxiety, the presence of an external time cue could help children manage uncertainty and regulate emotional activation before the task.

Similar to its potential role in emotional regulation, the visual timer may also support executive functions by offloading some of the cognitive demands associated with time monitoring. The timer may act as an external support for temporal processing, reducing the burden on executive functions and allowing students to better regulate their cognitive resources.

Difficulties in executive functions can lead to decreased on-task behaviors ([39]) including fidgeting, daydreaming, or engaging in activities unrelated to the task. It is thus hypothesized that the visual timer may promote on-task behaviors by supporting executive function. By making the passage of time more salient and predictable, the visual timer may create a more structured and supportive environment that fosters on-task behaviors and reduces off-task behaviors, such as fidgeting or motor impulsivity. Additionally, the timer may help to structure the task into smaller, more manageable intervals, enhancing the students’ sense of progress.

### 1.4. Aim of the Study

Building on previous findings, this study investigates the effects of using a visual timer on elementary school students’ performance, anticipatory anxiety levels, and behaviors during a mathematics assessment. The first objective of this study is to replicate the findings of [28] ([28]) regarding the positive effects of visual timers on anxiety and performance in an exam context. Specifically, we hypothesize that: (1) pupils’ mathematical performance will be higher in the visual timer condition compared to the timed assessment without a visual timer; and (2) the assessment with a visible timer will result in lower anticipatory evaluation anxiety compared to the evaluation without a visible timer. This design allows us to capture anticipatory anxiety, provides insight into how the visual timer influences children’s initial emotional regulation and readiness to perform.

Furthermore, this study expands upon previous research by investigating how children interact with the Time-Timer and to what extent they utilize it, while also analyzing the timer’s influence on their behavior throughout the task. This is an aspect that, to our knowledge, has not yet been investigated. We hypothesize that the visual timer condition will improve on-task behaviors with (3) fewer inattentive behaviors and (4) reduced motor instability compared to the no-timer condition. As an exploratory measure, we will also analyze how often children look at the Time-Timer to gain insight into their engagement with it. Finally, we will analyze whether individual cognitive processes, specifically selective attention and executive functions, moderate the relationship between the use of visual timers and the outcomes mentioned above (performance, anxiety, behavior). We hypothesized that (5) the extent to which individuals benefit from the visual timer will be related to their underlying cognitive abilities, particularly selective attention and executive functions. Individuals with stronger attentional abilities and executive functions may be better equipped to utilize the timer effectively, leading to greater improvements in on-task behaviors and potentially academic performance.

To test our hypotheses, 44 children aged 7 to 9 will participate in a mathematical assessment under two counterbalanced conditions: with and without a visual timer. This age range, similar to [28] ([28]), was chosen because it represents a critical period marked by substantial maturation in explicit time judgment abilities ([14]; [16]; [27]), making it an ideal in which to study the effects of interventions targeting time perception.

## 2. Method

### 2.1. Study Design

This quantitative study, conducted in French schools, investigated the impact of visual timers on mathematical performance, test-related anxiety, and off-task behaviors among students aged 7 to 9 years during a timed assessment. We employed a within-subjects design with two assessment contexts: one without a timer and one with a visual timer. Each participant took two versions of a mathematics test, administered on two separate mornings, one week apart, and always at the same time.

### 2.2. Participants

The study was conducted with a sample of 44 students (24 boys and 20 girls) enrolled in two schools located France [precise location was blanked to ensure anonymity during the review process]. The students were from 3 different class groups (i.e., 2 CE2 classes and 1 CE1 class, which correspond to grades 3 and 2 in the US system). At the time of the study, they were aged between 7 years 1 month and 9 years 0 months (mean age: 8.1 years, standard deviation: 0.5) and all participants were native French speakers and followed a typical French curriculum. The children’s parents signed written informed consent for participation in this study. The study adhered to the Helsinki Declaration and received approval from (1) the Academic School for Continuing Education of the academy where the experiment took place, (2) the Department of National Education Services of the region, and (3) the national education inspector.

### 2.3. Instruments

To answer our research questions, we relied on a set of standardized assessments, questionnaires, and observational measures targeting mathematics performance, assessment anxiety, attentional control, and executive functioning.

#### 2.3.1. Mathematics Assessment

The assessment materials were drawn directly from the open-access exercises provided in the study by [28] ([28]). These consisted of two mathematics tests adapted from Exercise 7 of the national mathematics assessment administered in September 2022 to all first-year elementary students (CE1) in the French education system. Each version included 14 multiple-choice calculations and 14 calculations requiring written answers. To ensure appropriate difficulty for second-year students (CE2), the latter authors also developed two versions of a more challenging task. Each student completed two versions of the assessment, with the difficulty level adjusted based on their class.

#### 2.3.2. Evaluation Anxiety Questionnaire (STA)

To measure children’s anticipatory anxiety in evaluative situations, we used the French version of the State Test Anxiety questionnaire (STA; [3]; [58]). This 6-item questionnaire, with responses on a 6-point Likert scale (1 = “not at all” to 6 = “very much”), has been validated in individuals aged 15 to 68 years. In our study, it demonstrated good internal consistency for our younger participants (Cronbach’s alpha = 0.80; McDonald’s omega = 0.80). The questionnaire was administered immediately after the instructions and before the beginning of the mathematics test, allowing us to assess children’s anticipatory state prior to task performance.

#### 2.3.3. Conners’ Questionnaire

The Conners’ scales (1969) were used to assess behaviors related to children’s attentional and executive functions. They are primarily used in clinical practice and research to identify children and adults with risk of Attention Deficit Hyperactivity Disorder (for a review, see [47]). The abbreviated version, consisting of 10 items with a 4-point Likert scale response format, is extremely short and easy to complete. It can be completed by any external observer (parent, teacher, tutor) and yields a unidimensional score for evaluating the intensity of hyperactive behaviors ([43]).

#### 2.3.4. Behavioral Observation Grid

During each mathematical assessment, student behaviors unrelated to the task were observed and counted. These were classified into two categories inspired by subtests of the NEPSY-II neuropsychological assessment ([37]) inattentive behaviors (e.g., turning head towards a distraction, looking out the window) and motor instability behaviors (e.g., getting up, fidgeting in the chair, changing pens). In addition, in the visual timer condition, the number of times the child looked at the timer was also recorded as a third dimension.

#### 2.3.5. Neuropsychological Tests

Two neuropsychological tests were taken from the TEA-Ch battery: Test of Everyday Attention for Children ([38]). The Sky Search test measures selective attention in the visual modality. The child is presented with a sheet containing 130 spaceships of 5 different types. The child’s task is to circle all pairs of identical spaceships as quickly as possible, ignoring distractors. Attentional quality is estimated by calculating the speed/accuracy ratio. The Doing Two Things at Once test measures sustained and divided attention in the visual and auditory modalities. The child must circle pairs of identical spaceships, while counting the laser-like gunshots heard during the task.

Two other tests were taken from the FEE battery: Child Executive Functions Assessment Battery ([50]). The Stroop test is a classic neuropsychological test ([53]) that assesses the ability to inhibit a dominant interfering response. In the version used in this study, two control conditions are proposed: color naming, which allows consideration of the child’s language skills, and the reading condition, which aims to reinforce the automaticity of reading color names. The condition of reading color names written in incongruent ink occurs in a third stage to measure the interference effect. The Trail Making Test is another widely used neuropsychological test ([49]) that assesses mental flexibility. In the FEE battery version, the child must first connect numbers in ascending order and then letters in alphabetical order. After these two control conditions, the child is asked to connect the numbers and letters alternately (1-A-2-B-3-C…), this condition requiring them to switch from one category to another with each item.

### 2.4. Procedure

After obtaining consent from the children and their parents, we collected data from January to April 2024. The mathematics assessment tasks were administered individually in a quiet place near the classroom; each child was exposed to two different contexts in two different days: without a timer and with a timer. A potential order effect was controlled by counterbalancing the two assessment contexts.

Before each test, standardized instructions were given orally “*Today, you will take a math test. On the first page, you need to find the result of each calculation and then circle it. When you have finished the first page, you turn the page: on the second page, you need to find the result of each calculation and then write it down. The goal is to go as fast as possible to get the most correct answers. If you don’t know, don’t circle anything and move on to the next calculation. There are a lot of calculations, it’s normal not to be able to do everything*.”

In the context without visual timer, last sentence was—“*This test is timed: you will only have 5 min. At the end of the 5 min, you will hear a buzzer, and you will have to put your pencil down, even if you have not finished*”. In the visual timer context, the last sentence was: “*This test is timed: you will only have 5 min. You will be able to see the time passing with the timer. At the end of the 5 min, you will hear a buzzer, and you will have to put your pencil down, even if you have not finished.*”

The evaluation of evaluation anxiety using the State-Trait Anxiety Inventory scale was carried out immediately after the instructions for the mathematics assessment task were given. This procedure allowed us to capture children’s emotional response to the upcoming evaluative situation rather than anxiety experienced during task execution.The individual administration of the neuropsychological tests was carried out following the two mathematics assessments, on a third day; the order of presentation of the tests was randomized.

### 2.5. Statistical Analysis

A power analysis was performed using G*Power (version 3.1) to determine the required sample size for a paired-samples t-test (two-tailed, α = 0.05, 1−β = 0.80). Estimating a medium effect size (Cohen’s dz = 0.50), the required sample size was estimated at *N* = 34. To account for potential variables, we planned to recruit at least 40 participants. Our final sample size was *N* = 44, exceeding the planned minimum.

Correct answers on the math test were counted for each child in both conditions (with and without a visual stopwatch). Descriptive statistics (means, standard deviations, skewness, and kurtosis) were calculated to summarize the data. All variables, except the number of errors, met the assumption of univariate normality, as indicated by skewness and kurtosis values between –2 and +2. Following these checks, repeated-measures ANOVAs were performed on correct responses, with Holm corrections (results were identical to Bonferroni corrections). Effect sizes were reported using η^2^. To further explore significant ANOVA effects, paired-samples t-tests were performed and Cohen’s *d* for paired t-tests, following standard recommendations.

Because the behavioral data (i.e., inattentive and motor instability behaviors) exhibited moderate floor effects and significant deviations from normality (e.g., kurtosis ≈ 4.45), a non-parametric approach was adopted to ensure the robustness and validity of the results. Therefore, non-parametric tests were prioritized for comparing the timer and no-timer conditions, and Spearman’s rho was used for the correlation analyses.

## 3. Results

### 3.1. Math Performance & Evaluation Anxiety

Table 1 descriptively presents the calculation scores rates and anxiety scores obtained by the students (n = 44) just after the instructions in the two experimental contexts (i.e., with a visual timer and without a visual timer).

While a slight increase in math scores was observed in the visual timer condition, a paired-samples t-test showed no significant difference in performance between the timer and no-timer conditions, *t*(44) = −0.44, *p* = 0.66, Cohen’s *d* = −0.067. Note that similar results were found when the analysis was run on the number of errors made by the children.

Notwithstanding, the analysis on anticipatory evaluation anxiety (STA scores) showed a significant effect of the experimental condition. The paired t-test confirmed that anxiety levels were significantly lower in the presence of a visual timer compared to the situation without a visible time (*t*(43) = 2.77, *p* = 0.008), Cohen’s *d* = 0.42.

### 3.2. Off-Task Behavior and Glances at the Time-Timer

Observation of behaviors during the task revealed a significant effect of the timer’s visibility on inattentive and motor instability behaviors (see Table 2). A Wilcoxon test confirmed that the number of inattentive behaviors was significantly lower in the presence of a timer (*W* = 55, *n* = 44, z = 2.803, *p* = 0.002, *r* = 0.42). A similar, but less pronounced, effect was observed for motor instability behaviors (*W* = 72, *n* = 44, z = 1.852, *p* = 0.03, *r* = 0.28).

Examining children’s engagement with the Time-Timer in the visible timer condition reveals heterogeneous usage patterns. Figure 1 illustrates the distribution of visual attention directed towards the Time-Timer which ranged from 0 to 12 glances (Mean = 4.41, SD = 3.26). Notably, two students did not look at the device, whereas a quarter of the participants (*n* = 10) engaged in frequent visual checks (more than 7 instances). The remaining students (*n* = 32) looked at the Time-Timer between one and seven times during the five-minute assessment.

### 3.3. Relationship with Neuropsychological Tests

Given the moderate floor effects and deviations from normality observed in inattentive and motor instability behaviors (kurtosis = 4.45), non-parametric correlation analyses (Spearman’s rho) were conducted to verify the robustness of the results.

A significant positive correlation was found between the decrease in inattentive behaviors in the presence of the timer and ADHD risk scores measured with the Conners’ questionnaire (ρ = 0.448, *p* = 0.013) indicating a medium-to-large effect size (See Table 3). This result indicates that children with higher ADHD risk scores benefited more from the presence of the visual timer. Other correlations were not significant.

To test the moderation hypothesis, we performed a linear regression analysis using the difference score for inattentive behaviors as the dependent variable and the Conners’ score as the predictor. This analysis confirmed a significant moderating effect of ADHD risk: children with higher Conners’ scores showed a significantly greater reduction in inattention when the visual timer was present. The model accounted for 15.4% of the variance in the difference score *F*(1, 28) = 5.10, *p* = 0.032, η^2^ = 0.15. Specifically, the Conners’ score significantly predicted the difference. No other significant moderating effects were observed for the remaining cognitive variables.

## 4. Discussion

This study aimed to investigate how the use of a visual timer impacted mathematical performance, anticipatory evaluation anxiety and on-task behaviors of 7- to 9-year-old students during a school assignment. We expected that using the timer would generally improve student performance, reduce anxiety, and decrease the number of off-task behaviors. We also postulated that these effects could be dependent upon the student’s executive functioning and attentional abilities.

First, our study successfully replicated the findings of [28] ([28]) regarding evaluation anxiety. Specifically, participants in the visible-timer condition demonstrated a significant reduction in anticipatory anxiety levels, measured immediately after the instructions and before task onset, relative to the condition without a visible timer. While both of our experimental conditions involved time limitations, it is the explicit presence of the timer in one condition that appears to have reduced anxiety upstream of the stain. While prior research often reports a link between time pressure and heightened anxiety in assessment situations ([8]; [40]), our results highlight that providing a clear indication of remaining time may, in fact, mitigate these effects. This finding thus offers a more nuanced perspective on the relationship between time pressure and anticipatory anxiety suggesting that the way time pressure is presented can significantly influence its impact. In the absence of a visible timer, individuals may experience heightened worry and rumination about the remaining time. This uncertainty likely consumes attentional resources, diverting them from the mathematical task. Conversely, the visible timer may lessen this cognitive burden by providing a clear and concrete representation of the time constraint.

Consequently, more attentional resources could be mobilized to engage in the task, particularly in mathematical problem solving, which was the focus of our experiment. Yet, our study found no significant impact of the timer’s visibility on participants’ performance. This contrasts with some prior research and warrants further consideration. Several factors may contribute to this lack of a discernible effect. One key difference lies in the testing environment: whereas the study that reported this effect employed group-administered assessments, our experiment involved individual testing sessions. This was necessary to precisely quantify gaze behavior toward the timer and other task-unrelated actions. The individual testing format, particularly the experimenter’s direct observation, may have optimized arousal and performance even in the condition without a visible timer. In this condition where participants were aware of the time constraint but lacked a visual cue, the presence of the experimenter might have served as a salient performance cue. A substantial body of research in social psychology underscores the significant influence of an observer’s presence on individual performance ([10]). Cottrell’s work further suggests that the observer’s perceived status can modulate this effect, with an “expert” observer tending to enhance motivation and performance. Moreover, the social control hypothesis ([24]) posits that the proximity of others and the direction of their gaze are important factors in social facilitation. Therefore, in our individual testing setting, the experimenter’s direct observation may have acted as a potent activating factor, potentially maximizing participants’ performance even when a timer was not visually present. This could have created a ceiling effect and reduced any potential for the timer to further enhance performance.

Although the timer’s visibility did not translate into measurable performance gains, its effects were particularly notable at the behavioral level, impacting how children managed their attention and activity during the task. Indeed, our results revealed a notable influence of the timer’s visibility on participants’ behavior during the task. Specifically, the presence of a visible timer corresponded with a reduction in both inattentive and motor instability behaviors. This suggests that when participants had a clear awareness of the remaining time, they exhibited greater focus and further controlled behavior. These findings align with the broader literature on the effects of time awareness on cognitive performance. For instance, research has shown that providing individuals with time-relevant information can enhance their self-regulation and task engagement (e.g., [36]). In the context of our study, the visible timer may have served as a regulatory cue, prompting participants to allocate their attention more effectively and minimize extraneous movements. This interpretation is also consistent with the attention control theory ([19]) previously exposed. By potentially mitigating uncertainty-related anxiety, the visible timer may have facilitated a more focused and deliberate approach to the task, resulting in fewer instances of inattentive and restless behaviors.

Beyond its role in reducing anticipatory anxiety, the impact of a visible timer could more broadly influence the ability to reduce cognitive load and free up additional attentional resources. This hypothesis seems to be supported by our results, particularly the finding that the effects of the visual timer on reducing off-task behaviors were more pronounced for children at higher risk for ADHD. This aligns with research suggesting that external aids and environmental modifications can be effective strategies for managing ADHD symptoms ([17]; [18]). Nonetheless, our study found limited evidence for a relationship between participants’ cognitive abilities, as measured by standard neuropsychological tests, and the observed variations in anxiety and behavior associated with the timer’s presence. If the effect of a visible timer had been solely attributable to attentional resources or executive functions, one would have anticipated replicating these effects on the other variables. A potential explanation for this inconsistency is the possibility of a non-linear relationship between these variables. Alternatively, it could be attributed to the Conners’ questionnaire’s primary focus on attentional and executive function impairment ([47]). Future research should specifically investigate the efficacy of visual timers as an intervention for children diagnosed with ADHD.

It should also be noted that other individual differences, not measured in this study, could also play a role. For instance, students’ perceived competence in mathematics (i.e., their self-efficacy; [2], [1]; [31]) or their general attitudes towards mathematics ([32]) might influence their anxiety between conditions and how they respond to the timer. Students with low self-efficacy might experience heightened anxiety even with a timer present, while those with positive attitudes towards school might be more receptive to its regulatory influence.

This observed variation in the frequency of glances at the Time-Timer, ranging from no glances to over twelve within the five-minute assessment, further complicates the interpretation of its impact. While the data indicates that many children did visually engage with the timer, suggesting its potential utility as a time management tool, the high frequency of glances observed in some participants (25%, n = 10, looked at it more than 7 times) raises questions about optimal usage patterns. It is possible that, for some children, particularly those in this age group (7–9 years old), the novelty of the Time-Timer led to excessive monitoring, potentially diverting attentional resources away from the task itself. This could be especially true if children are not accustomed to using such a tool for time management. The Time-Timer, although intended to reduce cognitive load, might have inadvertently increased it for some students who became overly focused on tracking the time. This highlights the importance of considering individual differences in how children interact with and adapt to new tools. Future research should investigate the potential benefits of providing explicit instruction and practice in using the Time-Timer, to determine whether familiarization and training can lead to more effective and less distracting usage patterns.

One limitation of this study is that the testing environment was individual. This condition, necessary for accurate behavioral observation, may have introduced a ceiling effect on performance, as the experimenter’s presence is a powerful predictor of performance, even in the absence of a visible timer. Future research should explore these effects in a more naturalistic classroom setting with group-administered tasks. Furthermore, the study’s focus on a relatively small sample of French students necessitates further investigation on a larger and more diverse scale to enhance generalizability. Furthermore, exploring the relationship between age, prior experience with visual timers, and the frequency of glances could shed light on developmental factors that influence the efficacy of this tool. To generalize these conclusions and broaden their scope, it would be worthwhile to expand the research by examining the effects of visual timers in groups and analyzing their impact over extended periods of use. This future research would provide crucial information on the ideal conditions for visual timers to optimize learning and engagement.

## 5. Conclusions

To conclude, this study highlights the multifaceted impact of visual timers, specifically the Time-Timer, on students’ cognitive, emotional, and behavioral dimensions during timed math assessments. While the timer did not directly enhance math performance in this particular context, its significant effect on reducing anticipatory anxiety before task onset and promoting on-task behaviors, particularly for students at higher risk for ADHD, underscores its potential as a valuable tool in educational settings. Ultimately, the Time-Timer may serve as a simple yet powerful tool for fostering a sense of control and reducing anxiety, allowing students to better focus on the task at hand and reach their full academic potential. However, the observed variability in usage patterns, particularly frequent glances, also highlights the need for further research on optimal implementation strategies.

## Figures and Tables

**Figure 1 ejihpe-15-00243-f001:**
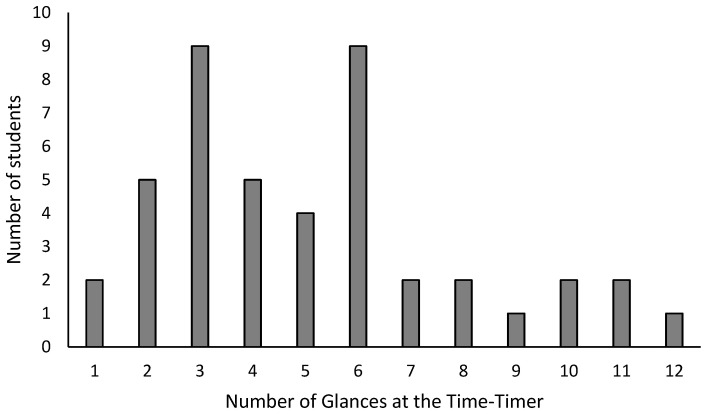
Frequency of Glances at the Time-Timer.

**Table 1 ejihpe-15-00243-t001:** Descriptive Statistics on the Number of Correct Answers on Math, as well as Anxiety scores (STA scale) as a function the assessment condition.

	Mean	SD	Kurtosis	Skewness
**Performance**				
No visual timer	17.32	6.82	−1.12	−0.28
Visual Timer	17.54	7.33	−1.22	−1.22
**Anxiety (STA)**				
No visual Timer	15.54	4.83	0.42	0.37
Visual Timer	13.84	4.64	0.23	0.55

**Table 2 ejihpe-15-00243-t002:** Descriptive Statistics on both Inattentive and Motor Instability as a function of the assessment condition (with and without a visible Time-Timer).

	Mean	SD	Kurtosis	Skewness
**Inattentive**				
No Timer	0.82	1.86	21.60	4.26
Visual Timer	0.48	1.42	31.35	5.28
**Motor Instability**				
No Timer	0.86	1.72	11.39	3.04
Visual Timer	0.52	1.02	8.05	2.60

**Table 3 ejihpe-15-00243-t003:** Correlations between cognitive measures (executive and attentional tests, ADHD risk) and the change in anxiety, inattentive behaviors, and motor instability between the no-timer and visible-timer conditions.

Variable	Anxiety Difference	Inattentive Behavior Difference	Instability Behavior Difference
**Executive Function**			
Inhibition (Stroop)	−0.06	0.02	−0.005
Attentional flexibility (Trail Making Test)	0.15	−10	−0.10
**Attention**			
Selective Attention (Sky search)	−0.02	−0.01	−0.16
Divided attention (Listen to two things at once)	0.19	−0.01	−0.01
Risk of ADHD (Conners’ test)	−0.21	**ρ = 0.448 ***	0.18

Note: * = *p* < 0.05. The correlation between ADHD risk (Conners’ test) and inattentive behavior difference was recomputed using Spearman’s rho to account for moderate non-normality (kurtosis = 4.45). The result remained significant (ρ = 0.448, *p* = 0.013).

## Data Availability

The data that support the findings of this study are available from the corresponding author upon reasonable request.

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
