# Peer review of "Time on Their Side: How Visual Timers Affect Anticipatory Anxiety, Performance, and On-Task Behavior in Elementary Math Assessments"

_ejihpe, 2025, doi:10.3390/ejihpe15120243_

Round 1

Reviewer 1 Report

Comments and Suggestions for Authors

This interesting study contributes meaningfully to understanding the effects of visual timers on children’s cognitive and emotional regulation during assessment tasks. The document is methodologically solid, clearly written, and theoretically supported by good literature.
Strengths:

  • Excellent theoretical framing linking time perception, attention, and anxiety.

  • Clear replication and expansion of previous findings.

  • Strong ethical and methodological transparency done within the study.

  • Balanced and insightful discussion.

Suggestions for improvement:

  • Clarify the generalizability aspect - Since testing occurred individually, results may not fully extend to classroom settings. Explicitly state this limitation in the final conclusion.
  • Add detail on effect sizes - Include η² or Cohen’s d values consistently in Results to strengthen interpretation.

  • Moderation analysis - Consider exploring whether attentional/executive variables act as moderators using regression models (even exploratory), instead of only correlations.

  • Reference list cleanup - Remove duplicate DOIs and standardised referencing (APA or journal format consistency).

  • Ethical statement—The rationale for not requiring a university IRB is clear, but it could be condensed for the reader's convenience.

  • Minor editing - Review typographical issues and ensure all tables/figures include captions explaining conditions fully.

Reviewer 2 Report

Comments and Suggestions for Authors

This submission is well-written and I was intrigued by the hypotheses and theoretical framework. Unfortunately, I cannot recommend your manuscript for publication as there is a fatal flaw in the design and a fatal flaw in the analyses.

The fatal flaw in the design is that (from page 7 lines 301 - 303)"The evaluation of anxiety using the State-Trait Anxiety inventory scale was carried out immediately after the instructions for the mathematics assessment task were given." To test whether condition (visual timer vs no visual timer) on anxiety due to math assessments should have been administered during or after the assessment. You state in the results that "anxiety levels were significantly lower in the presence of a visual timer compared to the situation without a visual timer." But what you found was anxiety levels were lower when the description of a visible timer was lower presented compared to description of no visible timer presented before the actual task. Although the inattentive behavior measures and instability behaviors were measure during the task, state anxiety was not. 

The fatal flaw in analysis is reported in section 3.3. Decreases in inattentive behaviors due to condition were greater (correlated with) for participants at risk for ADHD. Were there floor effects? Put differently, if participants had very low instances of inattentive behaviors, and were low risk of ADHD then it seems to me there might be some heteroskedasticity that could potentially lead to an inaccurate p value. Along with the fact that there appears to be a an issue of kurtosis which might have impacted the correlation coefficient is concerning.

I believe dropping the state anxiety analysis and addressing the correlation concerns would improve this manuscript, but then I am not sure you would have results to discuss other than the Off-task behaviors.

Round 2

Reviewer 2 Report

Comments and Suggestions for Authors

Thank you for addressing my concerns. Clarification of anticipatory anxiety as the variable of concern strengthens the paper. Thank you also for addressing my concern regarding the analysis.

Congratulations on a quality piece of research.